# Research

evolution

sponge-associated barnacle, larval settlement, adaptive evolution, host-driven evolution, symbiosis

**Author for correspondence:**
Benny Kwok Kan Chan
e-mail: chankk@gate.sinica.edu.tw

# Sponge symbiosis is facilitated by adaptive evolution of larval sensory and attachment structures in barnacles

Meng-Chen Yu[1,2], Niklas Dreyer[2,3,4,5], Gregory Aleksandrovich Kolbasov[6], Jens Thorvald Høeg[7] and Benny Kwok Kan Chan[2]

[1]Doctoral Degree Program in Marine Biotechnology, National Sun Yat-sen University and Academia Sinica, Kaohsiung 80424, Taiwan
[2]Biodiversity Research Center, Academia Sinica, Taipei 11529, Taiwan
[3]Department of Life Science, National Taiwan Normal University, Taipei, Taiwan
[4]Biodiversity Program, Taiwan International Graduate Program, Academia Sinica, Taipei, Taiwan
[5]Natural History Museum of Denmark, University of Copenhagen, Universitetsparken 15, DK-2100 Copenhagen, Denmark
[6]White Sea Biological Station, Biological Faculty of Moscow State University, Moscow 119899, Russia
[7]Department of Biology, Marine Biological Section, University of Copenhagen, Universitetsparken 4, DK-2100 Copenhagen, Denmark

M-CY, 0000-0001-9390-8090; ND, 0000-0002-1391-1642; GAK, 0000-0002-3762-1834; JTH, 0000-0002-3596-1691; BKKC, 0000-0001-9479-024X

Symbiotic relations and range of host usage are prominent in coral reefs and crucial to the stability of such systems. In order to explain how symbiotic relations are established and evolve, we used sponge-associated barnacles to ask three questions. (1) Does larval settlement on sponge hosts require novel adaptations facilitating symbiosis? (2) How do larvae settle and start life on their hosts? (3) How has this remarkable symbiotic lifestyle involving many barnacle species evolved? We found that the larvae (cyprids) of sponge-associated barnacles show a remarkably high level of interspecific variation compared with other barnacles. We document that variation in larval attachment devices are specifically related to properties of the surface on which they attach and metamorphose. Mapping of the larval and sponge surface features onto a molecular-based phylogeny showed that sponge symbiosis evolved separately at least three times within barnacles, with the same adaptive features being found in all larvae irrespective of phylogenetic relatedness. Furthermore, the metamorphosis of two species proceeded very differently, with one species remaining superficially on the host and developing a set of white calcareous structures, the other embedding itself into the live host tissue almost immediately after settlement. We argue that such a high degree of evolutionary flexibility of barnacle larvae played an important role in the successful evolution of complex symbiotic relationships in both coral reefs and other marine systems.

## 1. Introduction

Marine sponges (Porifera) are key players in coral reef ecology and can support a high diversity of symbiotic life (mutualism, parasitism and commensalism) [1–3]. The relationship between microbes and their sponge hosts has been widely explored [4–6]. Additionally, symbiotic lifestyles with sponges are widespread in invertebrate animals and can exhibit an array of remarkable specializations. This is seen in, for example, the sponge-dwelling shrimps, which have evolved novel mating strategies and eusociality [7]. Yet, little is known about the biology of sessile sponge-associated invertebrates. To establish symbiosis, the larvae of these organisms must first attach and then successfully metamorphose into a permanently sessile form embedded within the live tissue of their sponge host, but the severe difficulties in culturing and maintaining the

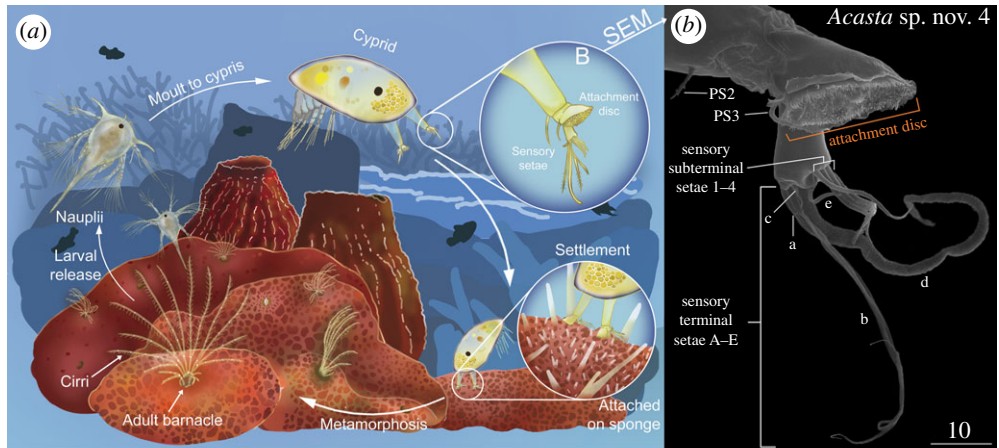

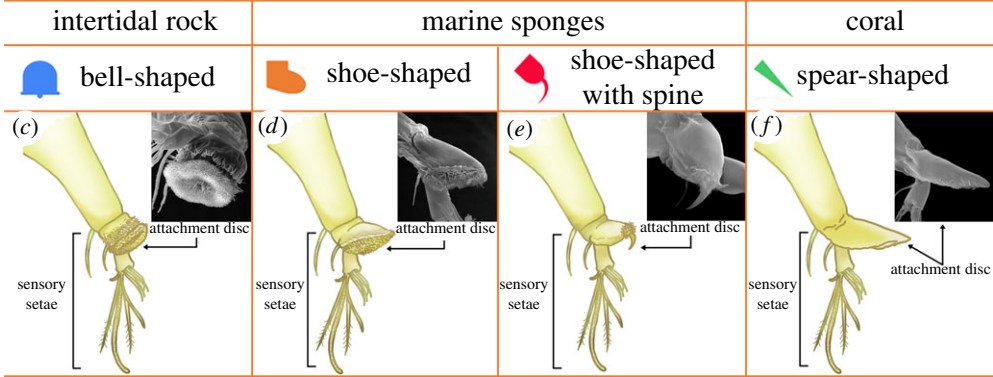

**Figure 1.** Life cycle of sponge-associated barnacles and variations in attachment disc structure linked to habitat adaption. (*a*) Life cycle of sponge barnacle. (*b*) SEM of cypris antennule structure of *Acasta* sp. nov. 4, showing the principal features of the attachment disc and sensory setae. (*c*) Bell-shaped attachment disc of *Capitulum mitella*. (*d*) Shoe-shaped attachment disc of *Euacasta dofleini*. (*e*) Shoe-shaped with spine attachment disc of *Acasta* sp. nov. 1. (*f*) Spear-shaped attachment disc of *Darwiniella angularis*. Scale bar in μm. (Online version in colour.)

relevant stages have until now prevented studies of how these tasks are accomplished.

Larvae settling on marine sponges must face a series of striking challenges. They must both avoid the dense layer of spiky spicules on the surface in order to attach (figure 1*a*) and also tolerate toxins and immobilizing anesthetics substances released by the hosts to deter fouling by symbiont larvae [8–10].

Barnacles offer an appealing avenue for illuminating how invertebrate larvae are capable of settling on and metamorphosing embedded in sponges. They inhabit not only an array of abiotic surfaces but are also symbiotic on a range of live substrata, including sharks and whales, jellyfish, corals and marine sponges [11–13]. This has resulted in elaborate adult morphologies, lifestyles and reproductive strategies [12,14–22].

Sponge-associated barnacles have their body embedded within the host tissue, and for suspension feeding they extend their cirri through a hole on the sponge surface (figure 1*a*) [19–22]. Their locations on sponges are not random, as they mostly occur on the inhalant side, which experience a stronger current for food capture (electronic supplementary material, video S1), implying that their locations are accurately determined by the settling larvae (figure 1*a*; electronic supplementary material, video S2) [19–22]. Previous studies proposed that the relationship between sponge-associated barnacles and their hosts is mutualist: barnacles can obtain trophic advantages and gain protection from the sponges, while barnacles can in turn strengthen

the skeleton of their hosts and also enhance sponge trophic intake [21]. There are, however, studies suggesting the barnacle and sponge relationship as being commensalism, such that barnacles obtain protection from host sponges and/or further reduce investment to shell physical armor [20,22], while the host neither suffers nor gains by housing the barnacles inside. Until now, the precise relationship between barnacles and their host sponge is uncertain. In this study, we consider that sponge-associated barnacles are symbiotic (living together) with their hosts without further evaluating their specific relationships due to lack of evidence at this moment.

Irrespective of adult biology, all barnacles share a unique larval stage, the cyprid, the sole purpose of which is to locate, explore and settle on a suitable substratum (figure 1). This is done using a pair of extendable antennules, equipped with an array of sensory and attachment devices, by first walking over the surface in an exploratory manner and finally attaching irreversibly by cement secretion (electronic supplementary material, video S2) [16,23–26]. Settlement of barnacles represents a unique decision, since it is irreversible whether on a bare rock in the intertidal zone or as a symbiont in the skin or tissue of a whale, crab, sponge or coral [24]. The settlement and metamorphosis of whale- and coral-associated barnacles have been recently studied [27–30]. The physics, chemistry and ecology of the substrata entail consequences not only for the attachment process itself but also for the subsequent metamorphosis and adult life. As an example, coral-associated barnacles have cyprids with attachment structures

uniquely modified to a spear shape for penetrating into the coral host [28].

Sponges not only differ from other host organisms used by barnacles but are also biologically very diverse among themselves, particularly in the morphology and density of their spicules [3] and in the sheer number of metabolites produced [31]. But until now, there are no observations how larvae of sponge-associated invertebrates settle and metamorphose on their hosts. In the present study, we successfully cultured a large number of sponge barnacle larvae and allowed them to settle and metamorphose (electronic supplementary material, videos S3 and S4). We particularly ask the following questions. (1) Have larval settlement on sponge hosts resulted in the evolution of novel structural adaptations not seen in other symbiotic species, such as in larval sensory or attachment structures? (2) Has sponge symbiosis entailed adaptations in either settlement behavior or metamorphosis? Using a molecular phylogeny, we first show that sponge barnacle symbiosis evolved homoplastically several times. By tracing the characters of both larva and the surfaces of their sponge hosts, we then examine whether these independent lineages lead to similar or different structural adaptations in the barnacle larvae.

## 2. Material and methods

Sponge barnacles and host sponges were collected in Taiwan, South Korea and Japan by scuba diving at depths of 3–20 m. The methods used here for sample collection, species identification (morphological and molecular approaches), scanning electron microscope (SEM) preparation and aquaculture of sponge barnacles and their hosts are described in electronic supplementary material, paragraph 1a–c and an article published in *Journal of Crustacean Biology* [19].

The parameters measured included maximum spicule length (average of 100 largest spicules per specimen), spicule content (spicule weight/total weight in %) and spicule density (spicule weight, mg/volume, cm$^3$) were estimated by measuring spicules weight and triple measures for each sample.

To compare variations in sponge texture characteristics, one-way analysis of variance (ANOVA) was conducted to examine variations in spicule density, content and maximum spicule length among host groups. Variations in parameters in sponge texture were subsequently analysed using multivariate analysis (PRIMER 6, Plymouth Routines in Multivariate Analysis) [32]. Before analyses, the data were square-root transformed to reduce the degree of differences among variables [32], and Euclidean distance was used for similarity matrix calculation. Non-metric multidimensional scaling (nMDS) was conducted to generate the two-dimensional plots of the sponge parameters between sponge species from each of the four types of cyprid antennules. We used analysis of similarity (ANOSIM) to test for differences in spicule parameters among sponge host groups.

Five molecular markers were analysed, namely, the mitochondrial 12S, 16S rRNA and cytochrome c oxidase subunit I (COI) genes, nuclear 18S rRNA gene and histone 3 (H3) [33–36]. We followed the conditions of the polymerase chain reaction (PCR) as described in references [19,33]. Direct sequencing of the purified PCR products was performed using the ABI 3730XL Genetic Analyzer with BigDye terminator cycle sequencing reagents (Applied Biosystems, Foster City, CA, USA). Sequences were assembled, edited and aligned in Geneious v. 7.1.4. [37]. Bayesian inference (BI) and maximum-likelihood (ML) analyses were conducted to reconstruct the phylogenetic relationships. We used jModeltest 2.1.7 [38] and

ModelFinder [39] to determine the best-fit evolutionary model for each partition. BI and ML analysis were conducted using MrBayes v.3.2.1 [40] and IQtree v.1.6.8 [41], respectively.

We adopted a BI approach to reconstruct the evolutionary history of larval morphological traits in sponge barnacles. We use the shape of attachment disc on the third segments for ancestral state reconstruction, using RASP 3.0 [42]. States of the characters were unordered and equally weighted. The analysis was repeated for all major nodes in our molecular phylogeny. The parameters of molecular phylogenetic analyses are described in electronic supplementary material, paragraph 1d and table S1c,d.

## 3. Results

### (a) Sensory and attachment structures in sponge barnacle cypris larvae

Figure 2 shows SEM micrographs of antennules in cyprids of sponge barnacles (figure 2a–h) and the surfaces of their particular hosts (figure 2m–p) with *in situ* photographs (figure 2i–l). We reared larvae of 18 species of sponge barnacles from the first naupliar instar to the cypris stage. Our SEM-based analysis of the antennular attachment structures revealed that the cyprids fall into four morphologically distinct groups (figure 2). These groups differed principally in the antennular parts that are in direct contact with the substrata during settlement (i.e. the third segment carrying the attachment disc), whereas other parts were virtually identical. It is especially notable that the SEM analysis revealed that cyprids of all groups had similar arrays of sensory setae on the fourth antennular segments (figures 1b and 2a–d), which are considered of primary importance in selecting the settlement site [23,43]. In all 18 species, the third antennular segments were shoe-shaped in lateral view (figure 2a–d), but other structural features enabled us to classify them into four groups. The differences between the groups concerned the density and distribution of minute villi covering the disc and the presence of hook-shaped devices (shoe-shaped with spine).

**Group 1.** (Even villus distribution; 10 species). The cyprids have a relatively high density of evenly distributed villi on the attachment disc, with longer villi towards the disc perimeter (figure 2e). In both this and the remaining three groups, the fourth segment resembles that found in other barnacles (thoracican) cyprids in having five terminally sited setae: two long setae A and B, a short and inconspicuous seta-C, an ornamented, aesthetasc-like seta-D and a simple and slender seta-E. In addition, this segment carries four long and subterminally sited setae (figure 1b).

**Group 2.** (Clustered villi; 6 species). The cyprids differed distinctly from the other groups in the density and disposition of the villi, which occur in scattered groups, separated by a distinct system of interconnecting grooves (figure 2b,f).

**Group 3.** (Bifurcated villi; 1 species). The cyprids differed from those in groups 1 and 2 by having a higher density of villi and in these being bifurcated, which again adds to the density. In addition, two antennular setae (PS2 and PS3), smooth-shaped in other types of sponge barnacle cyprids, here carry stout side branches, providing them with a rake-like or serrate appearance (figure 2c,g).

**Group 4.** (Disc with spine; 1 species). These cyprids exhibit by far the most extreme elaboration of the attachment disc

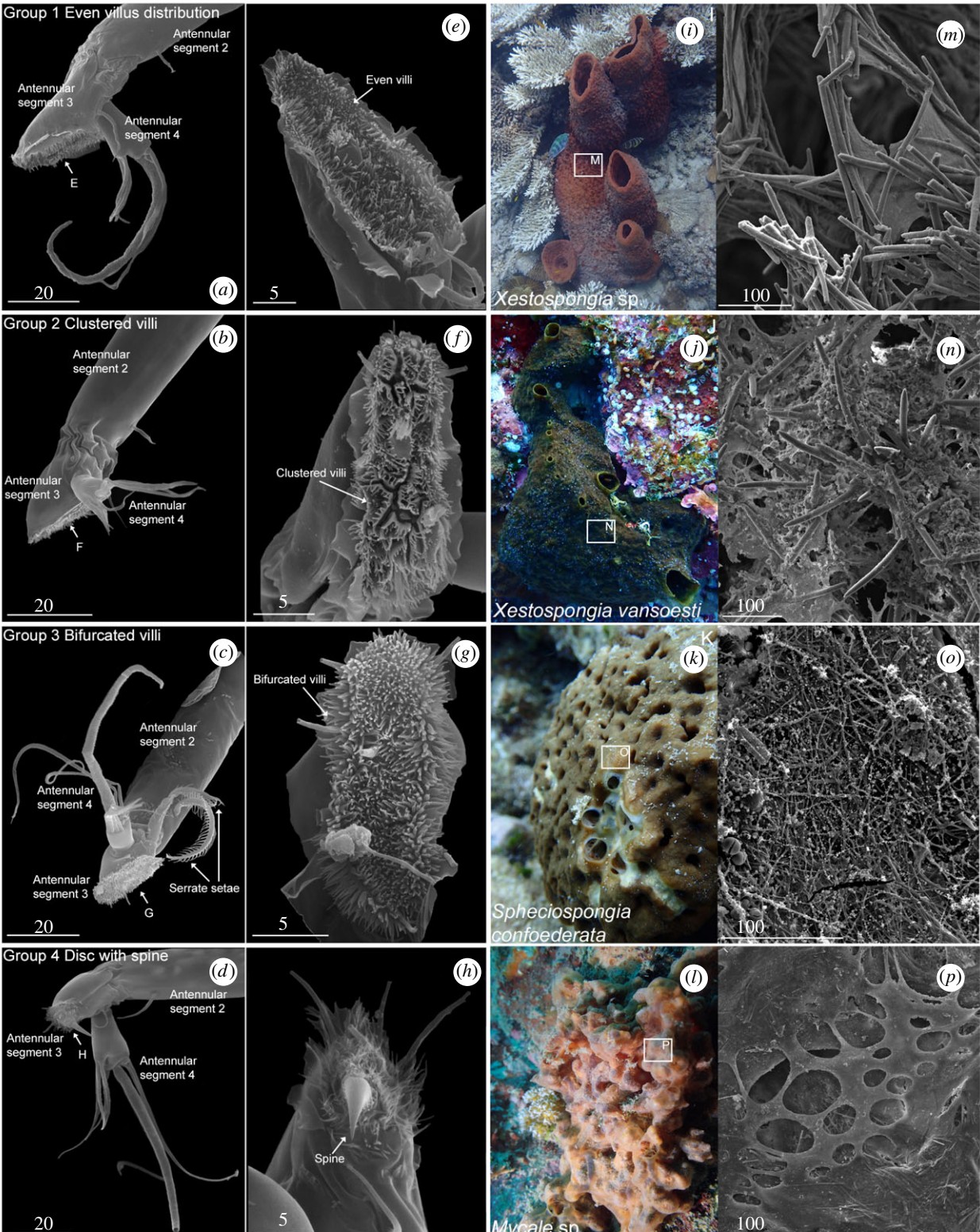

**Figure 2.** Antennular morphologies of four groups of sponge barnacles and their sponge hosts *in situ*, SEM images showing the shapes and structures of antennular third segments and attachment discs (AD), and sponge surface structures. (*a–d*) Side views of cypris antennules showing (*a–c*) shoe-shaped third segments and (*d*) shoe-shaped segment with spine. (*e–h*) Views of AD showing villi and other disc features, (*e*) even villi, (*f*) clustered villi, (*g*) bifurcated villi, and (*h*) thin villi with spine at centre of AD. (*i–l*) *In situ* photographs of sponge barnacle host sponges. (*m–p*) Surface of host sponges observed under SEM, (*m,n*) spiky spicule surfaces, (*o*) netted structure on the surface, and (*p*) conspicuous semicircular holes on the membranous surface. Scale bars in μm. (Online version in colour.)

armament yet seen in any barnacle [44,45]. Unlike groups 1–3, the attachment disc itself is very small and only sparsely covered with villi, which are here long and thin. The most defining characteristic was a prominent and slightly curved cuticular spine that extends from a depression in the attachment disc (figure 2*d,h*).

## (b) Multivariate analysis of cypris larval and sponge surface structures

The surface structure and spicule characteristics of the sponge host of four attachment disc groups were analysed (electronic supplementary material, table S2). Spicule density and

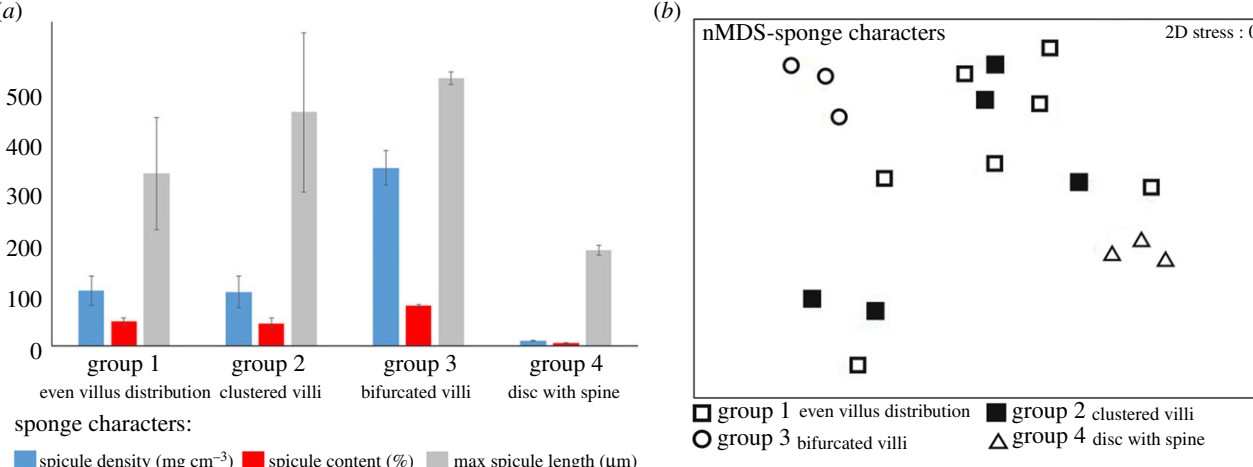

**Figure 3.** (*a*) The quantitative data of sponge spicular characters: spicule density (mg cm$^{-3}$), spicule content (%) and maximum spicule length (μm) based on the mean value of each sponge barnacle host group. (*b*) nMDS plot of sponge spicular characters of four sponge barnacle host groups. (Online version in colour.)

spicule content differed significantly among four types of host groups (figure 3*a*; one-way ANOVA, $F_{3,18} = 51.92$, $p < 0.01$ (spicule density); $F_{3,18} = 21.55$, $p < 0.01$ (spicule content)). Maximum spicule length did not differ significantly among host groups. Ad hoc pairwise SNK comparison indicates that hosts of groups 1 and 2 did not differ significantly in spicule density (119.58 and 116.27 mg cm$^{-3}$) or spicule content (52.72% and 48.26%; figures 2*m–n* and 3*a*). However, hosts of group 3 had by far the highest density and content of spicules (384.67 mg cm$^{-3}$ and 87.11%) and interlacing spongin fibres forming a netted structure on the surface (figures 2*o* and 3*a*). By contrast, hosts of group 4 had a very low density and content of spicules (9.60 mg cm$^{-3}$ and 5.42%) coupled with a very low spicule content of the sponge tissue (figure 3*a*). This rendered group 4 hosts with a virtually naked spongin surface with conspicuous semicircular holes (figure 2*p*).

Our multivariate analysis of sponge characteristics (maximum spicule length, spicule density and spicule contents) showed that the cyprid host groups formed three different clusters in the nMDS diagram (figure 3*b*; ANOSIM $R = 0.422$; $p \leq 0.05$). Pairwise comparisons indicates that hosts of groups 1 (even villus distribution) and 2 (clustered villi) cyprids could not be separated (figure 3*b*, open and filled squares), but hosts of group 3 (bifurcated villi) (figure 3*b*, circles) and group 4 (disc with spine) (figure 3*b*, triangles) were significantly different ($p < 0.05$; figure 3*b*).

## (c) Metamorphosis of sponge barnacle larvae

Prior to irreversible settlement, all cyprids used the antennules to walk, up to several hours, in an exploratory search over the sponge surface, much like other barnacles [25]. There are no obvious variations in this behaviour on the sponge surfaces between the two species examined. In some free-living barnacles, the cyprids are known to settle on the surface of the culture dish even in the presence of their natural substrata [46,47], but in our trials, the cyprids of sponge-associated barnacle always prefer their potential sponge host. Once settled, the pattern of metamorphosis deviates between the two species studied. *Euacasta dofleini* (Krüger, 1911) completed the entire process while remaining

on the sponge surface (*E. dofleini*; figure 4*a1–a6*; electronic supplementary material, video S4), while *Membranobalanus brachialis* (Rosell, 1973) embedded itself deep into the sponge tissue (*M. brachialis*; figure 4*b1–b6*; electronic supplementary material, video S3). In *M. brachialis*, the settled cyprid burrows into the surface within the first day. Subsequently, the metamorphosing juvenile situates itself deeper in the sponge until the cirri starts beating around 3 days after first contact with the host. Due to being buried, the details of metamorphosis could not be followed further. Any attempt to visualize buried, metamorphosing specimens by *in vivo* dissection resulted in extensive secretion of mucus obscuring the barnacle. In *E. dofleini*, the cells in the anterior part of the body begins to compact between 0 and 15 h after settlement, while simultaneously the epidermis begins to separate from the cypris carapace (white arrows, figure 4), signalling the onset of the cyprid-juvenile metamorphic moult. After 1 day, the primordial shell plates become visible beneath the cypris carapace. The cyprid-juvenile moult is completed 2 days after settlement by shedding of the cypris carapace, disintegration of the compound eyes and by complete retraction of cyprid antennular muscles. Subsequently, the body rotates 90° to the vertical. Cirral beating and feeding begin around 2–3 days after settlement (red arrows, figure 4; electronic supplementary material, videos S3 and S4). In *E. dofleini*, the juvenile starts developing several white, calcareous structures extending from their bodies after roughly 6 days. Removing these individuals from the hosts with forceps revealed that they were not fully cemented and/or strongly adhered to the host surface. While first being exposed, juveniles of *E. dofleini* are eventually overgrown by the host tissue.

## (d) Does attachment disc morphology reflect phylogenetic relationships?

We used five DNA barcoding markers of which three are mitochondrial (12S, 16S and COI) and two are nuclear (18S and H3) to generate our phylogenetic tree for 18 species of sponge barnacles, 20 species of symbiotic and free-living barnacles (superorder Thoracica) and 4 outgroup species (superorders Acrothoracica and Rhizocephala). We only included species in our analyses for which all five markers

*Euacasta dofleini* (epibiotic settlement and metamorphosis)

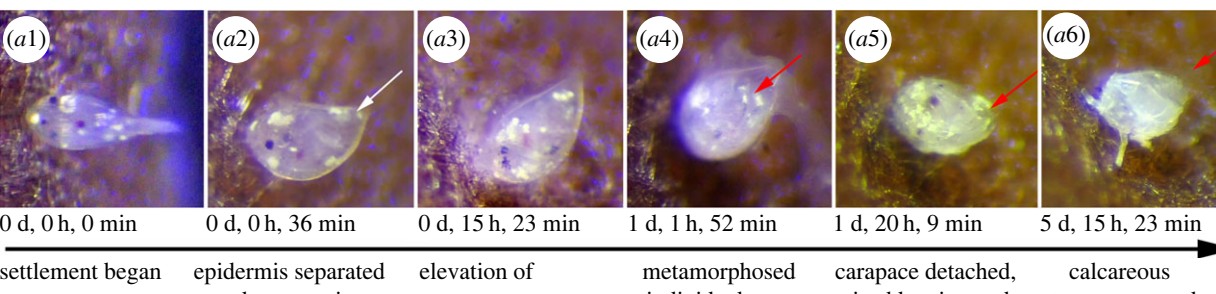

| (a1) | (a2) | (a3) | (a4) | (a5) | (a6) |
|---|---|---|---|---|---|
| 0 d, 0 h, 0 min | 0 d, 0 h, 36 min | 0 d, 15 h, 23 min | 1 d, 1 h, 52 min | 1 d, 20 h, 9 min | 5 d, 15 h, 23 min |
| settlement began | epidermis separated and compacting | elevation of carapace | metamorphosed individual seen underneath carapace | carapace detached, cirral beating and feeding | calcareous structure extend out from base |

*Membranobalanus brachialis* (embedding settlement and metamorphosis)

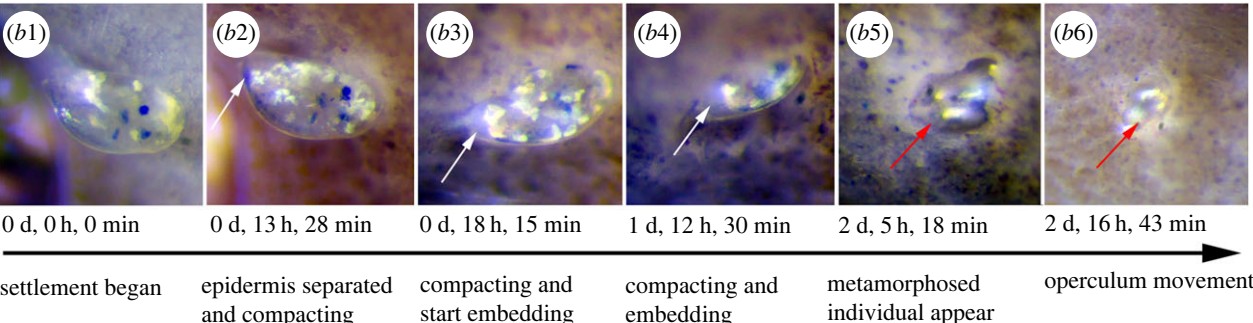

| (b1) | (b2) | (b3) | (b4) | (b5) | (b6) |
|---|---|---|---|---|---|
| 0 d, 0 h, 0 min | 0 d, 13 h, 28 min | 0 d, 18 h, 15 min | 1 d, 12 h, 30 min | 2 d, 5 h, 18 min | 2 d, 16 h, 43 min |
| settlement began | epidermis separated and compacting | compacting and start embedding | compacting and embedding | metamorphosed individual appear | operculum movement |

**Figure 4.** Epibiotic and embedding metamorphosis in the sponge barnacles *E. dofleini* (a1–a6) and *M. brachialis* (b1–b6), respectively. White arrows indicate the process, when the body begins to compact and the epidermis begins to separate from the cypris carapace; cirral beating and feeding are indicated by red arrows. (a1,b1) Settlement begins and the cypris body is lying close to the sponge surface, (a2,b2) epidermis separating and compacting, (a3) elevation of carapace, (b3 and b4) compacting and embedding, (a4,b5) metamorphosed individual, (a5) early juvenile, carapace has detached, start of cirral beating and feeding, and (a6) mature juvenile has developed calcareous structures extending out from the base. (b6) operculum movement. See electronic supplementary material for a video of the complete metamorphosis sequences. (Online version in colour.)

and cypris morphologies are available. For the phylogenies based on the concatenated dataset of all five markers, both the BI and ML topologies were similar (electronic supplementary material, figures S1 and S2). Figure 5 shows the ML topology with added Bayesian posterior probabilities (values below branches). The tree topology agrees with previous studies [33,35,48].

Our tree (figure 5) showed that the sponge-associated barnacles form a polyphyletic group and evolved independently at least three times from nonepibiotic forms (i.e. 'free-living' barnacles). The first group of sponge barnacles branches off as the sister group to the stony- and gorgonian-coral barnacles, together forming a monophyletic group (node A). The second group of sponge-inhabiting barnacles (*Acasta spongites* (Poli, 1791) and *Acasta cyathus* Darwin, 1854) branches off within a clade that also comprises the hydrocoral-inhabiting *Megabalanus ajax* (Darwin, 1854) and the rock-inhabiting *Amphibalanus amphitrite* (Darwin, 1854) (node B). This entire group is again sister group to the third clade of sponge barnacles (node C).

All free-living and nonsymbiotic species almost exclusively exhibited bell-shaped third antennular segments. All species settling on stony corals have secondarily evolved a spear-shaped third segment for host tissue penetration [28]. Similarly, all sponge-associated barnacle cyprids have through at least three separate lineages evolved a shoe-shaped third antennular segment from ancestors with a bell-shaped third segment. One lineage first evolved such a shoe-shaped segment, and this again gave rise to sponge barnacles with this morphology, while a second branch again evolved into coral barnacles with a spear-shaped segment.

## 4. Discussion

It is as yet unknown how tiny larvae of sponge symbiotic species can locate and attach to their potential sponge hosts. A somewhat comparative situation has now been studied in coral-associated barnacles, where the settling larva, the cyprid, must negotiate the nematocyst defenses of the future host [27,28]. Marine sponges have comparable defenses in the form of a layer of sharp spicules and their capability of excreting immobilizing anesthetics, which serve to deter settlement of fouling organisms. Thus, in both corals and sponges, the larvae of a symbiotic organism face serious challenges in order to settle and establish themselves.

We have for the first time used an approach where features in larval morphology (the attachment disc) and characteristics of the settlement substratum are jointly mapped onto a molecular phylogeny to test if they are coevolutionary linked. We suggest that the attachment discs of sponge-associated barnacles likely evolved in response to differences among their hosts. Our study shows that the ancestor to all acorn (Balanomorpha) barnacles, which inhabited the rocky intertidal, have a bell-shaped attachment disc. Subsequently, this evolved convergently several times into spear- or shoe-shaped discs, possibly as an adaptation to becoming epibiotic on corals and sponges. In these epibiotic species, other features of the cyprid antennules also seem to vary in association with differences among their hosts, culminating in the uniquely modified spine found in cyprids of one species. Similar patterns can be observed in the barnacle *Conopea* which grows on gorgonian corals. *Conopea granulata*

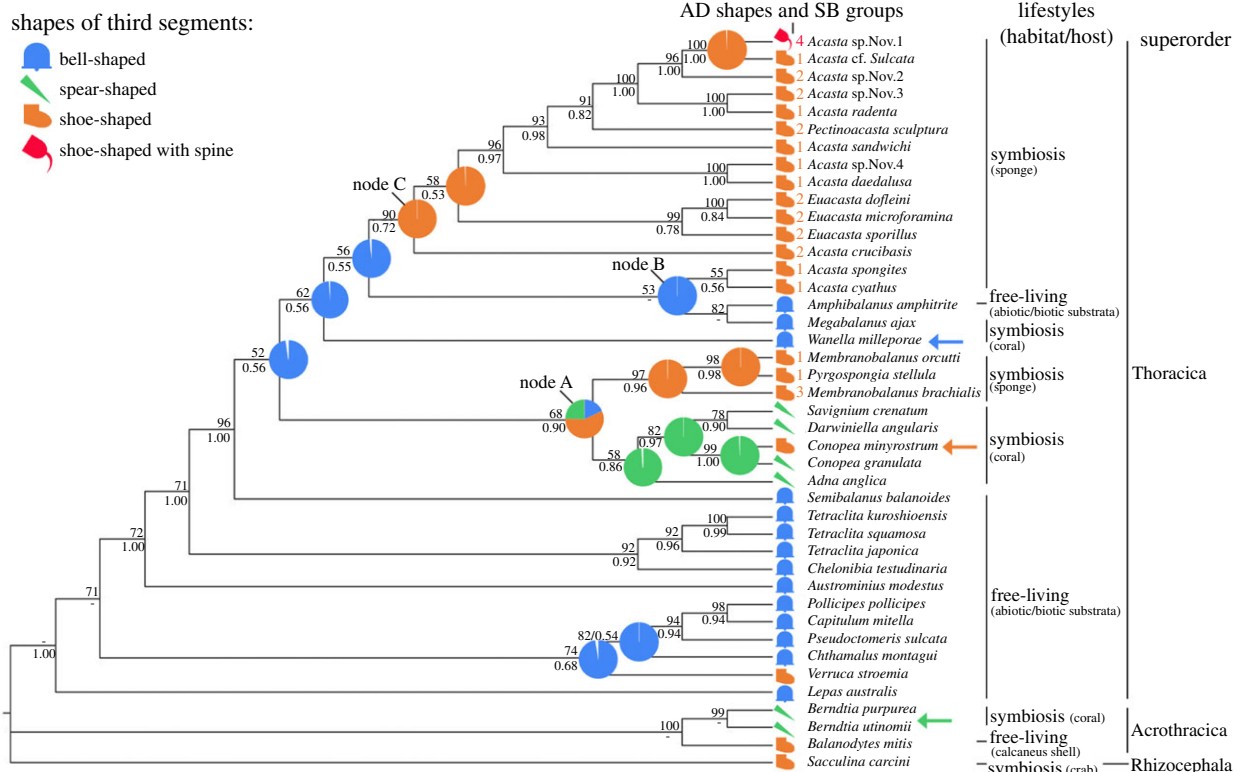

**Figure 5.** Phylogenetic relationships of superorders Thoracica, Acrothracica and Rhizocephala based on ML topology with reconstructed ancestral states of the shape of attachment discs (AD). The Bayesian posterior probabilities (PP; below branch) and ML support measures are indicated at nodes (— means ≤50). Ancestral state reconstruction is based on BI and implemented in RASP, estimated PP of alternative ancestral states are indicated by pie charts on nodes. Symbols next to the species names showing the shape of AD with sponge barnacle group numbers. Lifestyles, habitats or hosts of barnacles also are indicated and next to species names. Symbols and colours represent four major shapes: blue bell-, green spear-, orange shoe-shaped and red shoe-shaped with spine. (Online version in colour.)

(Hiro, 1937) grows on sea fans and its cyprid has a spear-shaped attachment disc, similar to most coral barnacles. In contrast, *Conopea minyrostrum* Van Syoc, Carrison-Stone, Madrona & Williams, 2014 is found on sea whips, which have spicule-covered surface just as found in sponges, and its cyprid has shoe-shaped antennules (orange arrow, figure 5). We therefore conclude that a key element in barnacle settlement, the cypris antennule, is indeed evolutionarily flexible when species invade new and biologically different habitats.

Although other remarkable transformations in cypris antennular function evolved among advanced parasitic forms [30,49,50], our group 4 species exhibits the most extreme modification known from any barnacle. Here, a large spine protrudes from the center of the attachment disc, and the only reasonable explanation is that it serves as an anchoring device that can be plunged into the soft host tissues. If so, this would be similar to the purely mechanical attachment (no cement secretion) seen in larvae of the closest relatives to the Cirripedia, the Ascothoracida, which are parasites of corals or echinoderms [51,52]. These examples strongly suggest fascinating and adaptive flexibility in exactly the elements directly involved in the settlement process even if the general aspects of the cypris larvae remain conservative [23,44,45].

Our results suggest that also the pattern of cuticular villi on the attachment disc may have evolved to counteract the surface properties of the host. The villi on the attachment disc are believed to play a key part in temporary and permanent adhesion of barnacle cyprids, but although studied with advanced techniques such as atomic force microscopy their

function remain largely unexplained [53–55]. Our study is one of the few to have focused on differences in these villi between species, and the distinct differences documented here underline their putative role. Interestingly, the group 4 cyprid has the lowest villus density among the species studied here, suggesting that the conventional chemical mechanisms used for temporary and permanent attachment may be of less importance [43,54].

Metamorphosis of sponge barnacles is almost entirely unknown, yet the adult stages are invariably covered by the host tissue [19–22]. Although we do not uncover the internal changes of tissues and organs during metamorphosis here, we show that the metamorphosis leading to the adult stages occur in very different ways in two sponge barnacle species, which also have different shell structures (electronic supplementary material, figure S4). One species, *E. dofleini*, has fully calcareous shells and remain on the surface during the entire process, while the other species, *M. brachialis*, has membranous basis and need to rapidly burrow itself into the live host tissue almost immediately after permanent settlement. Burrowing into sponges for metamorphosis may entail the benefit that sponge tissue can protect its weak membranous basis from attack by predators. We cannot rule out that this embedding is the result of antennular secretion that dissolves the sponge tissue as suspected in some parasitic barnacle cyprids [50], but emphasize that this needs experimental backing. In *E. dofleini*, the juvenile stages start developing several white and calcareous structures extending from their bodies around 6 days post settlement. The function of these calcareous extensions remains unknown, but we observed that the juveniles were not strongly cemented to

the host at this stage and could thus easily be removed. As metamorphosis progresses, the number of calcareous structures clearly increase as the barnacle gradually becomes surrounded by the host (figure 4*a*6; electronic supplementary material, figure S3), and they insert between the spongin and spicules of the latter. Given the exposed nature of such a mode of attachment and subsequent metamorphosis, we hypothesize that these structures serve the purpose of anchoring the juvenile further to the host before it is completely embedded.

In *M. brachialis*, the juvenile never develops such structures but is instead entirely embedded within the host by the completion of metamorphosis. These remarkable variations in the mode of host establishment and metamorphosis highlight the need for further studies on how sponge barnacles interact with their hosts during juvenile–adult metamorphosis.

## 5. Conclusion

We showed that the symbiotic association with marine sponges on coral reefs are driven and facilitated by flexible and adaptive evolution of larval attachment structures. The attachment device and metamorphic patterns of the larvae matched the surface properties of their hosts, indicating that such structures and processes can be highly host-driven. Our mapping of larval characteristics onto a molecular phylogeny documented both multiple origins of barnacle-sponge symbiosis and remarkably convergent adaptational changes of larval structures used for host attachment.

Data accessibility. All new data are available in the manuscript and electronic supplementary material.

Authors' contributions. All authors designed the project. M.-C.Y. and B.K.K.C. collected and analysed the data. N.D. and J.T.H. assisted in data analyses. All the authors wrote the manuscript.

Competing interests. We declare we have no competing interests.

Funding. This study was supported by the Crustacean Society Fellowship (2019) to M.-C.Y.; Russian Foundation for Basic Research grants no. 17-54-52006 MNT_a and 18-04-00624 A to G.A.K.; Danish Agency for Independent Research grant no. DFF-7014-00058 to J.T.H. and Senior Investigator Award of Academia Sinica AS-IA-105-L03 to B.K.K.C.

Acknowledgements. We thank P. C. Tsai, Y. F. Tsao, W. P. Hsieh and J. C. W. Liu (Academia Sinica) for assistance. We are grateful to S. P. Yu (Academia Sinica) for drawing figure 1*a,c–f*. We thank Kim Won (Seoul National University, Korea), T. Metzaki (Kuroshio Institute, Japan) and Yoko Nozawa (Academia Sinica) for assisting and arranging field work in Korea and Japan. M.-C.Y. is supported by doctoral program marine biotechnology (DDPMB) in National Sun Yat-sen University and Academia Sinica. N. D. acknowledges travel support by the Villum foundation and Dr Jorgen Olesen. N.D. is jointly supported by the Taiwan International Graduate Program (TIGP) and the Natural History Museum of Denmark. J.T.H. thanks Mr Bruno Hundsen for assistance.

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
