## [Reviewer comments · Proceedings of the Royal Society B: Biological Sciences]

Review History

RSPB-2019-2816.R0 (Original submission)

Review form: Reviewer 1 (Skarlatos G. Dedos)

Recommendation

Major revision is needed (please make suggestions in comments)

Scientific importance: Is the manuscript an original and important contribution to its field?

Good

General interest: Is the paper of sufficient general interest?

Acceptable

Quality of the paper: Is the overall quality of the paper suitable?

Poor

Is the length of the paper justified?

No

Should the paper be seen by a specialist statistical reviewer?

Yes

Do you have any concerns about statistical analyses in this paper? If so, please specify them explicitly in your report.

Yes

It is a condition of publication that authors make their supporting data, code and materials available - either as supplementary material or hosted in an external repository. Please rate, if applicable, the supporting data on the following criteria.

Is it accessible?

No

Is it clear?

No

Is it adequate?

No

Do you have any ethical concerns with this paper?

No

Comments to the Author

Reviewer Report on the manuscript titled: "Sponge symbiosis is facilitated by adaptive evolution of larval sensory and attachment structures in barnacles" by M-C Yu et al. submitted to Proceedings B, Manuscript ID RSPB-2019-2816

General comment

The manuscript by M-C Yu et al. is an interesting paper, well-written and concise, but, sadly, these are the only positive aspects of the paper.

This manuscript suffers tremendously from lack of proper reporting and so it is only guesswork how the authors came up with their conclusions! The authors give us in only 35 lines all the materials and methods that are critical for any reader to understand their results, when, in fact, they should have written at least 350 lines of materials and methods. I understand that Proceedings B requests for short and concise manuscripts, a practice I, personally, find irrational, but the authors have given a whole new meaning to a short and concise manuscript: inconceivable.

For example:

They state: "Scanning Electron Microscope (SEM) preparation and aquaculture of sponge barnacle and their host described in reference 34" when they should have written a whole page on this method instead of directing the reader to another of their paper. This is the highly criticized practice of self-citation (see News Feature on Nature 19/8/2019).

Another example:

They state: "Five molecular markers were analysed, namely the mitochondrial 12S, 16S rRNA and cytochrome c oxidase subunit I (COI) genes, nuclear 18S rRNA gene and histone 3 (H3)" and nowhere can one find this data or the alignment file or even what was aligned!

And another example:

They state: "BI and ML analysis were conducted using MrBayes v.3.2.1 [43] and IQtree v.1.6.8 [44], respectively" and nowhere can one find the parameters used to generate their phylogenetic tree using such highly-complicated software!

And all these inadequate reporting could have been remedied with a supplementary file where all the materials and methods were written in minute details.

As it stands, their manuscript can not be and will never be replicated by any other researcher!

Major Comments

- 1) The authors call the cyprid stage of barnacles a larval stage, an irksome wording when in fact, the cyprid stage is the equivalent of a nymphal stage brought about when a metamorphosis from a nauplius stage to the cyprid stage takes place.
- 2) They state: "SEM analysis revealed no differences between the sensory setae on the fourth antennular segments (figures 1b, 2a–d), which are considered of primary importance in selecting the settlement site" and one can not help but wonder: Has not this ring them any bells about how accurate their results are?
- 3) Figure 5 is impossible to understand since the authors show a cladogram that has no outgroup to root the tree, a prerequisite for a Bayesian analysis. Moreover, the Bayesian probability scores are very low at Node A and Node B, undermining their conclusions, and, in some branches, it is reported as a dash (-)! Furthermore, why don't they show us the BI and ML trees side by side separately as supplementary files?
- 4) They describe 4 groups of cyprids where Group 3 and 4 are represented by a single species each, and one can not help wonder whether their results would be entirely different if more species were included in Groups 3 and 4.

Minor comments

- 1) The images on Figure 4 are of too poor a quality to offer any support for their data. Either replace or direct the reader to the supplementary .mp4 files.
- 2) Figure 5 suffers from proper reporting since individual BI and ML trees must be shown side by side and alignment files must be provided as supplementary information.
- 3) They mention (ancestral) states of characters without giving any indication how these were generated.
- 4) They state: "The sponge hosts used in the present study can be categorised into groups based on their surface structure and spicule characteristics". Did they make an effort to identify the sponge species and see whether they belong to the same or different class or order?

Skarlatos G. Dedos

Review form: Reviewer 2

Recommendation

Accept as is

Scientific importance: Is the manuscript an original and important contribution to its field?

Excellent

General interest: Is the paper of sufficient general interest?

Good

Quality of the paper: Is the overall quality of the paper suitable?

Excellent

Is the length of the paper justified?

Yes

Should the paper be seen by a specialist statistical reviewer?

No

Do you have any concerns about statistical analyses in this paper? If so, please specify them explicitly in your report.

No

It is a condition of publication that authors make their supporting data, code and materials available - either as supplementary material or hosted in an external repository. Please rate, if applicable, the supporting data on the following criteria.

Is it accessible?

Yes

Is it clear?

Yes

Is it adequate?

Yes

Do you have any ethical concerns with this paper?

No

Comments to the Author

Studies of larval settlement mechanisms have been in vogue since before the 1950's and many papers demonstrate the responses of larvae to chemical sensory cues. Settlement adaptations of specialized symbiont larvae have been studied relatively little. This paper is a breath of fresh air. Its careful morphological analysis of the appendages used for sensing settlement cues and attachment demonstrate adaptations heretofore unimagined and point toward mechanisms that should be considered in other symbiotic relationships. Moreover, the use of molecular tools and cladograms give insight into the generality of the patterns observed. I liked this paper very much.

Review form: Reviewer 3

Recommendation

Major revision is needed (please make suggestions in comments)

Scientific importance: Is the manuscript an original and important contribution to its field?

Good

General interest: Is the paper of sufficient general interest?

Good

Quality of the paper: Is the overall quality of the paper suitable?

Acceptable

Is the length of the paper justified?

No

Should the paper be seen by a specialist statistical reviewer?

No

Do you have any concerns about statistical analyses in this paper? If so, please specify them explicitly in your report.

No

It is a condition of publication that authors make their supporting data, code and materials available - either as supplementary material or hosted in an external repository. Please rate, if applicable, the supporting data on the following criteria.

Is it accessible?

Yes

Is it clear?

Yes

Is it adequate?

Yes

Do you have any ethical concerns with this paper?

No

Comments to the Author

This is an interesting approach to studying barnacle settlement morphology by substrata and whether the barnacle was free-living or symbiotic. The findings support the hypothesis that a high degree of evolutionary flexibility in barnacle attachment structures facilitated the evolution of complex symbiotic relationships between them and sponge hosts.

The paper is strong in its approach, analysis and findings. The beautiful SEM images, great coding of the phylogenetic tree improves readability and understanding of data. The writing style and lack of an organized and impactful discussion is the main weakness to the paper and leaves the reader without context for the work. With a stronger discussion, this will be a great paper that contributes to our understanding of barnacle settlement strategies and evolution of symbiotic associations.

Points for consideration in the discussion:

- described major differences in villi in the third antennular segment, but there is no conjecture of the purpose of the villi in relation to the spicules/needles on the sponge.
- Any differences observed in terms of the exploratory behavior of cyprids with different antennular morphology?
- Any idea how exactly the embedding into sponge works? The antennular anchoring is explained but how do baseplates become embedded versus the epibiotic ones?
- Parallels are described with the timing and progression of settlement in epibiotic *Amphibalanus* amphitrite – any idea about adhesive content or timing of delivery with embedding settlers?

I have tried to highlight some of the important points that should be highlighted around which the discussion should be structured.

Line 29 ask:

Line 116a one-way analysis of variance (ANOVA)

Line 150SEM-based analysis

Line 244there should be one period after barnacles

Line 252what was the exception?

Lines 261-264 awkward

Lines 267-268 good topic sentence

Lines 285-287 important finding

Lines 299-302 interesting point

Lines 322-324 although the cement gland was not studied here, given how central it is to settlement and metamorphosis, is there anything that can be inferred from the different antennule morphology or known in embedding versus epibiotic settlers?

Line 327 “sport” is an odd word choice

Line 333-335 interesting point

Lines 345-348 seems like a major point to expand.
Lines 349-353 weak ending to the discussion

Decision letter (RSPB-2019-2816.R0)

06-Jan-2020

Dear Ms Yu:

I am writing to inform you that your manuscript RSPB-2019-2816 entitled "Sponge symbiosis is facilitated by adaptive evolution of larval sensory and attachment structures in barnacles" has, in its current form, been rejected for publication in Proceedings B.

This action has been taken on the advice of referees, who have recommended that substantial revisions are necessary. With this in mind we would be happy to consider a resubmission, provided the comments of the referees are fully addressed. However please note that this is not a provisional acceptance.

Sincerely,
Dr Daniel Costa
<mailto:proceedingsb@royalsociety.org>

Associate Editor
Board Member: 1
Comments to Author:

I think that this paper could be suitable for publication in PRSB after some substantial revision and improvement. The data and observations here are of highest quality and fit the journal scope, but the writing of the paper requires substantial re-organisation and improvement.

I would ask the authors to please try to ignore the tone of the comments by reviewer 1, which are deliberately discouraging to the point of being unprofessional. However, Reviewer 1, and also

Reviewer 3, make some important comments that will help you to improve the paper. If you are able to take their advice and restructure the paper I'm sure this will make a fine contribution to our journal. If you have any questions please feel free to contact me directly j.sigwart@qub.ac.uk

Julia Sigwart

Reviewer(s)' Comments to Author:

Referee: 1

Comments to the Author(s)

Reviewer Report on the manuscript titled: "Sponge symbiosis is facilitated by adaptive evolution of larval sensory and attachment structures in barnacles" by M-C Yu et al. submitted to Proceedings B, Manuscript ID RSPB-2019-2816

General comment

The manuscript by M-C Yu et al. is an interesting paper, well-written and concise, but, sadly, these are the only positive aspects of the paper.

This manuscript suffers tremendously from lack of proper reporting and so it is only guesswork how the authors came up with their conclusions! The authors give us in only 35 lines all the materials and methods that are critical for any reader to understand their results, when, in fact, they should have written at least 350 lines of materials and methods. I understand that Proceedings B requests for short and concise manuscripts, a practice I, personally, find irrational, but the authors have given a whole new meaning to a short and concise manuscript: inconceivable.

For example:

They state: "Scanning Electron Microscope (SEM) preparation and aquaculture of sponge barnacle and their host described in reference 34" when they should have written a whole page on this method instead of directing the reader to another of their paper. This is the highly criticized practice of self-citation (see News Feature on Nature 19/8/2019).

Another example:

They state: "Five molecular markers were analysed, namely the mitochondrial 12S, 16S rRNA and cytochrome c oxidase subunit I (COI) genes, nuclear 18S rRNA gene and histone 3 (H3)" and nowhere can one find this data or the alignment file or even what was aligned!

And another example:

They state: "BI and ML analysis were conducted using MrBayes v.3.2.1 [43] and IQtree v.1.6.8 [44], respectively" and nowhere can one find the parameters used to generate their phylogenetic tree using such highly-complicated software!

And all these inadequate reporting could have been remedied with a supplementary file where all the materials and methods were written in minute details.

As it stands, their manuscript can not be and will never be replicated by any other researcher!

Major Comments

- 1) The authors call the cyprid stage of barnacles a larval stage, an irksome wording when in fact, the cyprid stage is the equivalent of a nymphal stage brought about when a metamorphosis from a nauplius stage to the cyprid stage takes place.
- 2) They state: "SEM analysis revealed no differences between the sensory setae on the fourth antennular segments (figures 1b, 2a–d), which are considered of primary importance in selecting the settlement site" and one can not help but wonder: Has not this ring them any bells about how accurate their results are?
- 3) Figure 5 is impossible to understand since the authors show a cladogram that has no outgroup to root the tree, a prerequisite for a Bayesian analysis. Moreover, the Bayesian probability scores

are very low at Node A and Node B, undermining their conclusions, and, in some branches, it is reported as a dash (-)! Furthermore, why don't they show us the BI and ML trees side by side separately as supplementary files?

4) They describe 4 groups of cyprids where Group 3 and 4 are represented by a single species each, and one can not help wonder whether their results would be entirely different if more species were included in Groups 3 and 4.

Minor comments

1) The images on Figure 4 are of too poor a quality to offer any support for their data. Either replace or direct the reader to the supplementary .mp4 files.

2) Figure 5 suffers from proper reporting since individual BI and ML trees must be shown side by side and alignment files must be provided as supplementary information.

3) They mention (ancestral) states of characters without giving any indication how these were generated.

4) They state: "The sponge hosts used in the present study can be categorised into groups based on their surface structure and spicule characteristics". Did they make an effort to identify the sponge species and see whether they belong to the same or different class or order?

Skarlatos G. Dedos

Referee: 2

Comments to the Author(s)

Studies of larval settlement mechanisms have been in vogue since before the 1950's and many papers demonstrate the responses of larvae to chemical sensory cues. Settlement adaptations of specialized symbiotic larvae have been studied relatively little. This paper is a breath of fresh air. Its careful morphological analysis of the appendages used for sensing settlement cues and attachment demonstrate adaptations heretofore unimagined and point toward mechanisms that should be considered in other symbiotic relationships. Moreover, the use of molecular tools and cladograms give insight into the generality of the patterns observed. I liked this paper very much.

Referee: 3

Comments to the Author(s)

This is an interesting approach to studying barnacle settlement morphology by substrata and whether the barnacle was free-living or symbiotic. The findings support the hypothesis that a high degree of evolutionary flexibility in barnacle attachment structures facilitated the evolution of complex symbiotic relationships between them and sponge hosts.

The paper is strong in its approach, analysis and findings. The beautiful SEM images, great coding of the phylogenetic tree improves readability and understanding of data. The writing style and lack of an organized and impactful discussion is the main weakness to the paper and leaves the reader without context for the work. With a stronger discussion, this will be a great paper that contributes to our understanding of barnacle settlement strategies and evolution of symbiotic associations.

Points for consideration in the discussion:

- described major differences in villi in the third antennular segment, but there is no conjecture of the purpose of the villi in relation to the spicules/needles on the sponge.
- Any differences observed in terms of the exploratory behavior of cyprids with different antennular morphology?
- Any idea how exactly the embedding into sponge works? The antennular anchoring is explained but how do baseplates become embedded versus the epibiotic ones?

- Parallels are described with the timing and progression of settlement in epibiotic Amphibalanus amphitrite – any idea about adhesive content or timing of delivery with embedding settlers?

I have tried to highlight some of the important points that should be highlighted around which the discussion should be structured.

Line 29 ask:

Line 116a one-way analysis of variance (ANOVA)

Line 150SEM-based analysis

Line 244there should be one period after barnacles

Line 252what was the exception?

Lines 261-264 awkward

Lines 267-268 good topic sentence

Lines 285-287 important finding

Lines 299-302 interesting point

Lines 322-324 although the cement gland was not studied here, given how central it is to settlement and metamorphosis, is there anything that can be inferred from the different antennule morphology or known in embedding versus epibiotic settlers?

Line 327 “sport” is an odd word choice

Line 333-335 interesting point

Lines 345-348 seems like a major point to expand.

Lines 349-353 weak ending to the discussion

Author's Response to Decision Letter for (RSPB-2019-2816.R0)

See Appendix A.

RSPB-2020-0300.R0

Review form: Reviewer 3

Recommendation

Accept with minor revision (please list in comments)

Scientific importance: Is the manuscript an original and important contribution to its field?

Good

General interest: Is the paper of sufficient general interest?

Good

Quality of the paper: Is the overall quality of the paper suitable?

Acceptable

Is the length of the paper justified?

Yes

Should the paper be seen by a specialist statistical reviewer?

No

Do you have any concerns about statistical analyses in this paper? If so, please specify them explicitly in your report.

No

It is a condition of publication that authors make their supporting data, code and materials available - either as supplementary material or hosted in an external repository. Please rate, if applicable, the supporting data on the following criteria.

Is it accessible?

Yes

Is it clear?

Yes

Is it adequate?

Yes

Do you have any ethical concerns with this paper?

No

Comments to the Author

The clarity and flow of the paper is much improved, although I have made several minor suggestions to improve the readability. Please consider incorporating these suggestions so that the major findings of this paper are more salient. This is very interesting and detailed work that is an important contribution to the field.

Review form: Reviewer 4

Recommendation

Accept with minor revision (please list in comments)

Scientific importance: Is the manuscript an original and important contribution to its field?

Excellent

General interest: Is the paper of sufficient general interest?

Good

Quality of the paper: Is the overall quality of the paper suitable?

Good

Is the length of the paper justified?

Yes

Should the paper be seen by a specialist statistical reviewer?

No

Do you have any concerns about statistical analyses in this paper? If so, please specify them explicitly in your report.

No

It is a condition of publication that authors make their supporting data, code and materials available - either as supplementary material or hosted in an external repository. Please rate, if applicable, the supporting data on the following criteria.

Is it accessible?

Yes

Is it clear?

Yes

Is it adequate?

Yes

Do you have any ethical concerns with this paper?

No

Comments to the Author

I found this to be an interesting, novel and well-constructed contribution to the literature on barnacle biology, and larval settlement ecology more broadly. I would urge the authors to have the paper proof-read, since the grammar and sentence structure, including mixed tenses etc., let it down. It is nice work and deserves to be seen in the best light.

I am reviewing the most recent version of the paper, of course, but it appears that the comments of previous reviewers (the reasonable ones, at least) have been addressed.

I have one significant query. The fact that the authors are so confident about their terminology, and that it was not raised in previous reviews, could suggest that this is my inadequate understanding, but I would ask for some clarity on use of the definition 'symbiont' or 'symbiosis' to describe the relationship between barnacles and their sponge hosts. This seems to be more of a parasitic relationship to me (what does the sponge get out of it?) or at least epi/endo-biotic. Lines 60-63 compounded my views on this. Surely cooperation is a hallmark of a symbiotic relationship, rather than challenging or deterring the other party? I ask for either a change in terminology or, in the introduction, a statement defining how the relationship is a symbiotic one.

Decision letter (RSPB-2020-0300.R0)

25-Mar-2020

Dear Ms Yu:

Your manuscript has now been peer reviewed and the reviews have been assessed by an Associate Editor. The reviewers' comments (not including confidential comments to the Editor) and the comments from the Associate Editor are included at the end of this email for your reference. As you will see, the reviewers and the Editors have raised some concerns with your manuscript and we would like to invite you to revise your manuscript to address them.

To submit your revision please log into <http://mc.manuscriptcentral.com/prsb> and enter your

Author Centre, where you will find your manuscript title listed under "Manuscripts with Decisions." Under "Actions", click on "Create a Revision". Your manuscript number has been appended to denote a revision.

Research ethics:

Use of animals and field studies:

All supplementary materials accompanying an accepted article will be treated as in their final form. They will be published alongside the paper on the journal website and posted on the online figshare repository. Files on figshare will be made available approximately one week before the

accompanying article so that the supplementary material can be attributed a unique DOI. Please try to submit all supplementary material as a single file.

Please submit a copy of your revised paper within three weeks. If we do not hear from you within this time your manuscript will be rejected. If you are unable to meet this deadline please let us know as soon as possible, as we may be able to grant a short extension.

Best wishes,
Dr Daniel Costa
mailto:proceedingsb@royalsociety.org

Associate Editor

Comments to Author:

Thank you for your additional revisions to the paper, I agree with the reviewers that this paper should be accepted after some additional minor revisions to incorporate the suggestions of both referees. Please pay particular attention to the revision or justification of the term "symbiosis" and also make sure that you carefully proof read the entire paper in its final form to remove grammatical mistakes that would otherwise distract the reader from your lovely results.

Reviewer(s)' Comments to Author:

Referee: 3

Comments to the Author(s).

The clarity and flow of the paper is much improved, although I have made several minor suggestions to improve the readability. Please consider incorporating these suggestions so that the major findings of this paper are more salient. This is very interesting and detailed work that is an important contribution to the field.

Referee: 4

Comments to the Author(s).

I found this to be an interesting, novel and well-constructed contribution to the literature on barnacle biology, and larval settlement ecology more broadly. I would urge the authors to have the paper proof-read, since the grammar and sentence structure, including mixed tenses etc., let it down. It is nice work and deserves to be seen in the best light.

I am reviewing the most recent version of the paper, of course, but it appears that the comments of previous reviewers (the reasonable ones, at least) have been addressed.

I have one significant query. The fact that the authors are so confident about their terminology, and that it was not raised in previous reviews, could suggest that this is my inadequate understanding, but I would ask for some clarity on use of the definition 'symbiont' or 'symbiosis' to describe the relationship between barnacles and their sponge hosts. This seems to be more of a parasitic relationship to me (what does the sponge get out of it?) or at least epi/endo-biotic. Lines 60-63 compounded my views on this. Surely cooperation is a hallmark of a symbiotic

relationship, rather than challenging or deterring the other party? I ask for either a change in terminology or, in the introduction, a statement defining how the relationship is a symbiotic one.

Author's Response to Decision Letter for (RSPB-2020-0300.R0)

See Appendix B.

Decision letter (RSPB-2020-0300.R1)

14-Apr-2020

Dear Ms Yu

I am pleased to inform you that your manuscript entitled "Sponge symbiosis is facilitated by adaptive evolution of larval sensory and attachment structures in barnacles" has been accepted for publication in Proceedings B.

Open Access

Paper charges

Sincerely,

Dr Daniel Costa
Editor, Proceedings B
mailto: proceedingsb@royalsociety.org

Associate Editor:
Board Member

Comments to Author:

Thank you for your revisions to this paper, I am happy to recommend it to be accepted for publication in PRSB

Julia Sigwart

Appendix A

February 11, 2020

Editor, Proceedings of the Royal Society B: Biological Sciences

Subject: Revision and resubmission of manuscript RSPB-2019-2816

Dear Editor,

Thank you for your email dated 06 January 2020, informing us to improve and resubmit our manuscript entitled "Sponge symbiosis is facilitated by adaptive evolution of larval sensory and attachment structures in barnacles." We have addressed all reviewers' comments and our specific responses were listed in the specific responses are as follows:

Associate Editor:

Comment: *I think that this paper could be suitable for publication in PRSB after some substantial revision and improvement. The data and observations here are of highest quality and fit the journal scope, but the writing of the paper requires substantial re-organisation and improvement.*

I would ask the authors to please try to ignore the tone of the comments by reviewer 1, which are deliberately discouraging to the point of being unprofessional. However, Reviewer 1, and also Reviewer 3, make some important comments that will help you to improve the paper. If you are able to take their advice and restructure the paper I'm sure this will make a fine contribution to our journal. If you have any questions, please feel free to contact me directly <j.sigwart@qub.ac.uk>.

Response: We thank the associate editor for encouraging and positive comments. We have taken the advice of reviewers to re-organise and restructure our manuscript in the revised version.

Referee 1:

General comment

Comment: *This manuscript suffers tremendously from lack of proper reporting and so it is only guesswork how the authors came up with their conclusions.*

Response: Thanks for your comments. We would disagree that our conclusions are purely guesswork. Our specific responses below explains our standpoints in details.

Comment: *The authors give us in only 35 lines all the materials and methods that are critical for any reader to understand their results, when, in fact, they should have written at least 350 lines of materials and methods. I understand that Proceedings B requests for short and concise manuscripts, a practice I, personally, find irrational, but the authors have given a whole new meaning to a short and concise manuscript: inconceivable.*

February 11, 2020

For example:

They state: “Scanning Electron Microscope (SEM) preparation and aquaculture of sponge barnacle and their host described in reference 34” when they should have written a whole page on this method instead of directing the reader to another of their paper. This is the highly criticized practice of self-citation (see News Feature on Nature 19/8/2019).

Another example:

They state: “Five molecular markers were analysed, namely the mitochondrial 12S, 16S rRNA and cytochrome c oxidase subunit I (COI) genes, nuclear 18S rRNA gene and histone 3 (H3)” and nowhere can one find this data or the alignment file or even what was aligned!

And another example:

They state: “BI and ML analysis were conducted using MrBayes v.3.2.1 [43] and IQtree v.1.6.8 [44], respectively” and nowhere can one find the parameters used to generate their phylogenetic tree using such highly-complicated software!

And all these inadequate reporting could have been remedied with a supplementary file where all the materials and methods were written in minute details.

As it stands, their manuscript cannot be and will never be replicated by any other researcher!

Response: Thank you for your suggestion. We tend to agree with some of these arguments. Materials and methods should be described in details in a manuscript. However, the page limitation of Proceedings B restricted our methods section. We consulted many other papers from the journal and concluded that ours was in the range of that published in the paper. To address the issue, we have, electronic supplementary materials, paragraph S1, table S1 and 2, added an elaborate description of all methods and GenBank accession numbers for each sequence used in this study. We referred to ourselves because the methods were indeed described in details in Yu et al. (2019) and therefore we see no need for a repetition. Although we have not submitted to Nature, we do not consider this as a case of critical self-citation as described in the News Feature 19/8/2019.

Yu M-C, KolbasovGA, Høeg JT, Chan BKK. 2019 Crustacean-sponge symbiosis: collecting and maintaining sponge-inhabiting barnacles (Cirripedia: Thoracica: Acastinae) for studies on host specificity and larval biology. *J.Crustac. Biol.* **39**, 522–532. (doi:10.1093/jcbiol/ruz025)

Major Comments

Comment: *The authors call the cyprid stage of barnacles a larval stage, an irksome wording when in fact, the cyprid stage is the equivalent of a nymphal stage brought*

February 11, 2020

about when a metamorphosis from a nauplius stage to the cyprid stage takes place.

Response: This critique revolves around terminology. This manuscript is not the right place to challenge the 150+ year old and well-established term “larva” when describing the cyprid. The nymphal stage is used to describe an immature life stage of insects before they reach their adult stage. A key feature is that the morphology of nymphs is almost entirely similar to that of the adult stage, i.e., the metamorphosis is hemimetabolous. With insect larvae the metamorphosis is holometabolous. The cyprid indeed undergoes a full metamorphosis in all barnacles, ranging to the extreme in the parasitic forms. The morphology of the cyprid is therefore drastically different from the succeeding juvenile and adult stages. Thus, cyprid can be truly be considered a true larval form and this is also the way it is treated in all cirripede and marine larval literature to this day.

Except the thecostracan cypris larvae, other crustaceans (see Malacostraca) also possess specialized larval stages following naupliar stages, e.g. zoea, antizoea, pseudozoea, mysid larva etc. (for more information read Martin et al., 2014).

Martin JW, Olesen J, Høeg JT. (eds.) 2014 Atlas of crustacean larvae. MD: Baltimore, Johns Hopkins University Press.

Comment: *They state: “SEM analysis revealed no differences between the sensory setae on the fourth antennular segments (figures 1b, 2a–d), which are considered of primary importance in selecting the settlement site” and one cannot help but wonder: Has not this ring them any bells about how accurate their results are?*

Response: Our main objective is to compare the morphological variation on the third segment (the attachment disc), because this is the structure that is used directly for substratum walking. It is in this segment that we find significant structural variation between cyprids of both sponge barnacles in particular and barnacles in general. We interpret this as adaptations to the different habitats and substrates used by the species. Remarkably, the fourth segment varies very little if at all in structure between species, even when they differ widely in habitat and structure. It always consists of four subterminal setae and five terminal, and the external morphology and number of these setae among species are extremely similar. We have revised the word ‘no differences’ as ‘similar’ in this sentence in page5 line157.

The function of the fourth segment is considered to be a critical sensor of the substrate and environmental factors (largely unknown), but it is difficult to observe the action of those sensory setae (although Maruzzo et al. (2012) did exactly that, in a study including one of us). The conclusions reached both here and in previous studies (largely by members of our group) is that the fourth segment can do its job of

February 11, 2020

chemo- and mechano-reception with little structural change, probably because the adaptation lies at the neuronal level. See data and conclusions in Al-Yahya et al. (2016) and Chan et al. (2017) where antennules from many species were compared using morphometric statistics. We elaborate on this here only because of the referee comment. These points have already been dealt with in other papers and there seems no need to repeat such discussion in our present MS.

Maruzzo D, Aldred N, Clare AS, Høeg JT. 2012 Metamorphosis in the cirripede crustacean *Balanus amphitrite*. *PLoS ONE* **7**, e37408. (doi:10.1371/journal.pone.0037408)

Al-Yahya H, Chen H-N, Chan BKK, Kado R, Høeg JT. 2016 Morphology of cyprid attachment organs compared across disparate barnacle taxa: does it relate to habitat? *Biol. Bull.* **231**, 120–129. (doi:10.1086/690092)

Chan BKK, Sari A, Høeg JT. 2017 Cirripede Cypris Antennules: How Much Structural Variation Exists Among Balanomorph Species from Hard-Bottom Habitats? *Biol. Bull.* **233**, 135–143. (doi:10.1086/695689)

Comment: *Figure 5 is impossible to understand since the authors show a cladogram that has no outgroup to root the tree, a prerequisite for a Bayesian analysis. Moreover, the Bayesian probability scores are very low at Node A and Node B, undermining their conclusions, and, in some branches, it is reported as a dash (-)! Furthermore, why don't they show us the BI and ML trees side by side separately as supplementary files?*

Response: Thank you for your suggestion. We have now highlighted outgroup used in page 7, line 245-246. They are the superorders Acrothracica and Rhizocephala, and this was also listed in the methods section (electronic supplementary materials, paragraph S1d, table S1).

Some of the bootstrap values on the tree have been revised and corrected due to typos, the value reported as a dash if value lower than 50 (see figure 5). We agree with the low Bayesian posterior probability (BI tree) and bootstrap values (ML tree). However, the reconstruction of the two trees are based on five genetic markers (12S, 16S, 18S, COI and H3, total length 2468bp), and the patterns addressed in our study are similar to the previous phylogenies reported in Chan et al. (2017) and Perez-Losada et al. (2008). Today, the resolution of multiple genes for barnacle phylogenetic reconstruction is still insufficient and face limitations. Increasing the length of sequences of course can provide more reliable means to consider the historical events (combined with fossil ages). We would like to keep the existing tree with BI and ML bootstrap value in the MS figure. But we provide the individual BI and ML trees in electronic supplementary materials, figures S1,2.

February 11, 2020

Chan BKK, Corbari L, Rodriguez Moreno PA, Tsang LM. 2017 Molecular phylogeny of the lower acorn barnacle families (Bathylasmatidae, Chionelasmatidae, Pachylasmatidae and Waikalasmatidae) (Cirripedia: Balanomorpha) with evidence for revisions in family classification. *Zool. J. Linn. Soc.* **180**, 542–555. (doi:10.1093/zoolinnean/zlw005)

Perez-Losada M, Harp M, Høeg JT, Achituv Y, Jones D, Watanabe H, Crandall KA. 2008 The tempo and mode of barnacle evolution. *Mol. Phylogenet. Evol.* **46**, 328–346. (doi:0.1016/j.ympev.2007.10.004)

Comment: *They describe 4 groups of cyprids where Group 3 and 4 are represented by a single species each, and one can not help wonder whether their results would be entirely different if more species were included in Groups 3 and 4.*

Response: Thank you for your comment. In our current collection, we have around 35 species of sponge barnacles, and we have successfully reared 18 species to the cypris stage. We face high technical difficulties in the successful culturing of the complete larval development, since the nauplii are feeding and experience high mortality rates. Since this study is the first on sponge barnacle cyprids (and includes many more species experimentally reared than in any previous study of barnacle larva), we believe that it is a sufficient starting point for understanding their settlement biology and morphology. We have mentioned in the discussion that more species should be collected.

Larvae from species with group 3 and group 4 morphologies have not been reported and studied before. Thus, this is the best result we have at the present and we believe that our main finding is clear: that common ancestry does not play a major role in the differences found in the antennular structures that interact directly with the hosts. We believe that it is of minor importance that we have but one species with an antennular hook or large spine. In fact, we argue that this is a significant, new and highly interesting structural feature. It was never seen before in any other barnacle cyprid and seems to relate to the host surface. It is a truly unique feature among barnacle cyprids and we consider it one of the most remarkable finding in our work.

Minor comments

Comment: *The images on Figure 4 are of too poor a quality to offer any support for their data. Either replace or direct the reader to the supplementary .mp4 files.*

Response: Thank you for your comment. The images of the metamorphosis patterns are derived from stereomicroscopes at very high magnification which has relatively shallow depth of view. This is a technical limitation because only using stereomicroscopes can allow observation of live individuals for the whole continued

February 11, 2020

metamorphosis patterns. At this moment, the quality of photos is enough to show the distinct stages of metamorphosis and two types of metamorphosis. We feel this figure is essential because we can clearly document two types of metamorphosis, which again has never been shown before.

Electronic supplementary materials, videos S1,2 show the entire epibiotic and embedding settlement process. This allows the readers to understand how cyprids walk, explore, sense and settle on their hosts. We recommend the reader to watch these videos for more information about the cypris settlement process.

Comment: *Figure 5 suffers from proper reporting since individual BI and ML trees must be shown side by side and alignment files must be provided as supplementary information.*

Response: Thank you for your suggestion. We put the ML and BI bootstrap values at each node. The BI and ML trees, alignment file and GenBank numbers, are provided as electronic supplementary materials, figures S1,2, file S1 and table S1.

Comment: *They mention (ancestral) states of characters without giving any indication how these were generated.*

Response: Thank you for your comment. We have added the method of reconstructing the ancestral state in phylogenetic analysis in electronic supplementary materials, paragraph S1d.

Comment: *They state: "The sponge hosts used in the present study can be categorised into groups based on their surface structure and spicule characteristics". Did they make an effort to identify the sponge species and see whether they belong to the same or different class or order?*

Response: Thank you for your suggestion. We have added the species name of host sponges in electronic supplementary materials, table S1. The sponge host usage of 18 barnacle species comprises 8 families in 5 orders, all belong to Class Demospongiae.

Referee 2:

Comment: *Studies of larval settlement mechanisms have been in vogue since before the 1950's and many papers demonstrate the responses of larvae to chemical sensory cues. Settlement adaptations of specialized symbiont larvae have been studied relatively little. This paper is a breath of fresh air. Its careful morphological analysis of the appendages used for sensing settlement cues and attachment demonstrate adaptations heretofore unimagined and point toward mechanisms that should be*

February 11, 2020

considered in other symbiotic relationships. Moreover, the use of molecular tools and cladograms give insight into the generality of the patterns observed. I liked this paper very much.

Response: We thank the reviewer for the encouragement and positive response.

Referee 3:

Comment: *This is an interesting approach to studying barnacle settlement morphology by substrata and whether the barnacle was free-living or symbiotic. The findings support the hypothesis that a high degree of evolutionary flexibility in barnacle attachment structures facilitated the evolution of complex symbiotic relationships between them and sponge hosts.*

The paper is strong in its approach, analysis and findings. The beautiful SEM images, great coding of the phylogenetic tree improves readability and understanding of data. The writing style and lack of an organized and impactful discussion is the main weakness to the paper and leaves the reader without context for the work. With a stronger discussion, this will be a great paper that contributes to our understanding of barnacle settlement strategies and evolution of symbiotic associations.

Response: We thank reviewer for positive comments and providing many important viewpoints for our manuscript. We have revised and re-organised the manuscript according to the suggestions.

Comment: *Described major differences in villi in the third antennular segment, but there is no conjecture of the purpose of the villi in relation to the spicules/needles on the sponge.*

Response: In the discussion (4th paragraph), we have a paragraph that states that villi density may relate to the sponge surface morphology. This is the only reasonable explanation with the present data at hand. The villi are pure cuticle and thus cannot have a sensing function. More likely, their morphology may optimize the attachment being structurally specialized to the host texture, and perhaps especially so during temporary reversible attachment during surface exploration. Recent papers (Phang et al., 2008; Walker, 1992; Yap et al., 2017) have addressed the mechanism of temporary attachment including the role of the villi but no thorough understanding has been achieved, even though both Atomic Force Microscopy and high speed video has been employed. Therefore, we refrain from undue further speculation. A future goal should be live observation of villi function on the sponges.

Phang IY, Aldred N, Clare AS, Vancso GJ. 2008 Towards a nanomechanical basis for temporary

February 11, 2020

adhesion in barnacle cyprids (*Semibalanus balanoides*). *J. R. Soc. Interface* **5**, 397–402. (doi:10.1098/rsif.2007.1209)

Walker G. 1992 Cirripedia. In *Microscopic anatomy of invertebrates* (ed FW Humes), pp. 249–311. New York, NY: Wiley-Liss Inc.

Yap FC, Wong W-L, Maule AG, Brennan GP, Chong VC, Lim LHS. 2017 First evidence for temporary and permanent adhesive systems in the stalked barnacle cyprid, *Octolasmisangulata*. *Sci. Rep.* **7**, 44980. (doi:10.1038/srep44980)

Comment: *Any differences observed in terms of the exploratory behavior of cyprids with different antennular morphology?*

Response: We have observed the exploratory behavior of cyprids in the two species which successfully metamorphosed on sponges. There are no obvious differences in the way these cyprids explored on the surfaces (see page 6, lines 214-215). We have commented on this in the manuscript now.

Comment: *Any idea how exactly the embedding into sponge works? The antennular anchoring is explained but how do baseplates become embedded versus the epibiotic ones?*

Response: The embedding process appears involve secretions from the attachment disc to digest or dissolve the sponge tissue, thus allowing the cyprids body to get into the sponges. A similar situation occur in parasitic barnacles and possibly in cetacean barnacles, as well as the acrothoracicans. *M. brachialis* has membranous basis and settled by embedding metamorphosis, in contrast to *E. dofleini*, which has a calcareous basis, and settles on sponge surfaces (see pages 9-10, lines 317-343). This has been commented on in the last section in the discussion, although not in great details because we have no real data on this.

Comment: *Parallels are described with the timing and progression of settlement in epibiotic Amphibalanus amphitrite – any idea about adhesive content or timing of delivery with embedding settlers?*

Response: At present, it is very difficult to observe how the settlement and metamorphosis occurs inside, because cutting out the sponges will stimulate mucus secretion from the hosts. This naturally makes it difficult to observe the physical process, particularly alive. Cutting away the sponge tissue would cause mucus production and would obscure any metamorphic process (see page7, lines 223-226). The adhesive material of sponge barnacles has not been sequenced, but we thank the reviewer for the comment and will make this an on-going project. It is, indeed, very interesting to see whether the materials themselves differ between substrata.

February 11, 2020

Comment: I have tried to highlight some of the important points that should be highlighted around which the discussion should be structured.

Line 29 ask:

Line 116 a one-way analysis of variance (ANOVA)

Line 150 SEM-based analysis

Line 244 there should be one period after barnacles

Line 252 what was the exception?

Lines 261-264 awkward

Lines 267-268 good topic sentence

Lines 285-287 important finding

Lines 299-302 interesting point

Lines 322-324 although the cement gland was not studied here, given how central it is to settlement and metamorphosis, is there anything that can be inferred from the different antennule morphology or known in embedding versus epibiotic settlers?

Line 327 "sport" is an odd word choice

Line 333-335 interesting point

Lines 345-348 seems like a major point to expand.

Lines 349-353 weak ending to the discussion

Response: Thanks for the suggestion on the improvement of the discussion. In the revised manuscript, we totally re-structured the discussion. The flow of discussion includes 1) adaptive evolution of cyprids attachment discs, 2) function of villi on attachment disc, 3) discussion on epibiotic and embedding settlement and metamorphosis. We hope this re-structured discussion can enhance the strength of the discussion.

We hope our resubmitted manuscript can be accepted for publication.

Sincerely,

Meng-Chen Yu, Niklas Dreyer, Gregory A. Kolbasov, Jens T. Høeg and Benny K.K. Chan

Appendix B

April 03, 2020

Editor, Proceedings of the Royal Society B: Biological Sciences

Subject: Revision of manuscript RSPB-2020-0300

Dear Editor,

Thank you for your email dated 25 March 2020, informing us to revise our manuscript entitled "Sponge symbiosis is facilitated by adaptive evolution of larval sensory and attachment structures in barnacles." We have addressed all reviewer' comments, as detailed below:

Associate Editor:

Comment: *Thank you for your additional revisions to the paper, I agree with the reviewers that this paper should be accepted after some additional minor revisions to incorporate the suggestions of both referees. Please pay particular attention to the revision or justification of the term "symbiosis" and also make sure that you carefully proof read the entire paper in its final form to remove grammatical mistakes that would otherwise distract the reader from your lovely results.*

Response: We thank the associate editor for encouraging and positive feedback. We have cited the references Kolbasov, 1993; Ilan et al., 1999; Magnino et al., 1999; Yu et al., 2019 to explain "why sponge barnacles are symbiotic?" We now also state our viewpoint of the relationship between sponge barnacles and their hosts in the introduction, page 3 lines 71-87. We stated "Sponge-associated barnacles have their body embedded within the host tissue, and for suspension feeding they extend their cirri through a hole on the sponge surface (figure 1a) [Yu et al., 2019; Kolbasov, 1993; Ilan et al., 1999; Magnino et al., 1999]. Their locations on sponges are not random, as they mostly occur on the inhalant side, which experience a stronger current for food capture (electronic supplementary material, video S1), implying that their locations are accurately determined by the settling larvae (figure 1a; electronic supplementary material, video S2) [Yu et al., 2019; Kolbasov, 1993; Ilan et al., 1999; Magnino et al., 1999]. Previous studies proposed the relationship between sponge-associated barnacles and the sponges as mutualism; barnacles can obtain trophic advantages and gain protection from host sponges. Barnacles inside sponges can strengthen the sponge skeleton and enhance trophic intake of host sponge [Magnino et al., 1999]. There are however, studies suggested the barnacle and sponge relationship as commensalism, i.e., that barnacles obtain protection from host sponge and/or further reduce investment to shell physical armor [Ilan et al., 1999 and Kolbasov, 1993] while the host neither suffers nor gains being housed the barnacles inside. Until now, the precise relationship between barnacles and their host sponge is uncertain. In this study, we consider the sponge associated barnacles are symbiotic (living together) with their hosts without further evaluating their specific relationships due to lack of evidence at this moment."

April 03, 2020

Kolbasov GA. 1993 Revision of the genus *Acasta* leach (cirripedia: balanoidea). Zool. J. Linn. Soc. 109, 395–427. (doi.org/10.1111/j.1096-3642.1993.tb00307.x)

Ilan M, Loya Y, Kolbasov GA, Brikner I. 1999 Sponge inhabiting barnacles from coral reefs. Mar. Biol. 133, 709–716. (doi:10.1007/s002270050)

Magnino G, Pronzato R, Sara' A, Gaino E. 1999 Fauna associated with the horny sponge *Anomoianthella lamella* Pulitzer-Finali & Pronzato, 1999 (lanthellidae, Demospongiae) from Papua-New Guinea. Ital. J. Zool. 66, 175–181. (doi:10.1080/11250009909356253)

Yu M-C, Kolbasov GA, Høeg JT, Chan BKK. 2019 Crustacean-sponge symbiosis: collecting and maintaining sponge-inhabiting barnacles (Cirripedia: Thoracica: Acastinae) for studies on host specificity and larval biology. J. Crustac. Biol. 39, 522–532. (doi:10.1093/jcbiol/ruz025)

Referee 3:

Comment: *The clarity and flow of the paper is much improved, although I have made several minor suggestions to improve the readability. Please consider incorporating these suggestions so that the major findings of this paper are more salient. This is very interesting and detailed work that is an important contribution to the field.*

Response: Thank you for positive comments and suggestions, which help us to improve the manuscript. We have addressed all suggestions in the revised manuscript.

Comment: *Minor suggestions to improve the clarity and flow of the paper:*

Lines 113. ...spicule length (average of 100 largest spicules per specimen)

Lines 117, 199, 352. Use characteristics instead of characters

Line 121. Insert “the” before data. Why were the data transformed? State which normality test was applied, if appropriate.

Lines 129-130. Eliminate this sentence and put the citations with the previous one.

Line 151. Hyphenate SEM-based. This describes the analysis.

Lines 162-164. “...evenly distributed... with longer villi towards the disc perimeter.”

Lines 170-171. “...cyprids differed in the density...”

Lines 181-182. Flip the sentence to read, “ The most defining characteristic was a prominent and slightly curved cuticular spine that extends from a depression in the attachment disc.”

Line 191. Replace “But” with “However,”

Lines 209-210. Add a comma after “walk” “hours” and “surface”. Change “as known from” to “much like”.

Line 213. Eliminate “frequently”. They don’t frequently settle, they settle once.

Lines 282-284. Change “demonstrate” to “suggest” or say “The data suggest that...”. Insert likely between barnacles and evolved and eliminate “adaptively”.

Lines 284-286. “... barnacles, which inhabited the rocky intertidal, have a bellshaped...”

April 03, 2020

Line 310. Consider saying “to counteract the” instead of “in correlation with”.

Line 315. Consider saying “... underlie their putative role”.

Line 335. Instead of “Dovetailing with this”, consider saying “As metamorphosis progresses,”

Line 342. Replace “already during” with “by the completion of”

Line 343. Insert “the” between “in” and “mode”.

Line 350. “metamorphic” is the adjective form to describe “patterns”.

Line 352. “host-driven”

Response: Thanks for the suggestions and comments. We have addressed all the editing suggestions in our manuscript.

Comment: *Line 121. Insert “the” before data. Why were the data transformed? State which normality test was applied, if appropriate.*

Response: We have inserted “the” before data. Our raw data with wide range values from different types of data including the maximum spicule length, spicule content and spicule density. Square root data transformation can reduce the degree of differences among variables, before entering the multivariate analysis. We have added this in page 4 lines 133-135.

Comment: *Lines 130-131. Consider saying “PCR conditions differed from ref 31 as follows:” to state how your methods differed.*

Response: We have rephrased the sentence as “We followed the conditions of the polymerase chain reaction (PCR) as described in references 19 and 32”. Reference 32 is the same as 31 (before revised).

In the original sentence, “we modified the conditions of the PCR...”, “modified” means that we may adjust the annealing temperature of PCR for each sample to increase the success rate of PCR.

Comment: *Line 156. Armament is an odd word choice here. You may want to save it for the discussion when you discuss the spear-like segment.*

Response: We have rephrased “armament” as “arrays” of the sensory setae in page 5 line 169.

Comment: *Line 214-215. This is unclear. Are you saying that they did not prefer the sponge to other substrates?*

Response: We have rephrased the sentences as “the cyprids of sponge-associated barnacle always prefer their potential sponge host” in page 7 lines 228-229. In our observation, sponge barnacle cyprids never settle on other substrata; they exclusively settle on their host

April 03, 2020

sponge.

Comment: *Line 263-264. Do you have a reference that this is used for host-tissue penetration or is this something observed in this study?*

Response: We have cited a reference (Liu et al., 2016) about the host-tissue penetration by coral-association barnacle cyprid in page 8 line 278.

Liu JCW, Høeg JT, Chan BBK. 2016 How do coral barnacles start their life in their hosts? Biol. Lett. 12, 20160124. (doi:10.1098/rsbl.2016.0124)

Comment: *Line 294. Should Madrona and Williams, 2014 be a parenthetical reference here?*

Response: It is binomial nomenclature of *Conopea minyrostrum* Van Syoc, Carrison-Stone, Madrona & Williams, 2014.

Comment: *Line 317. What do you mean by chemical mechanism here? It seems that you are discussing a biophysical mechanism with the villi and surface properties of the host?*

Response: We refer to the chemical mechanism being “cement secretion” during settlement used for temporary and permanent attachment. We have rephrased the sentence as “suggesting that the conventional chemical mechanisms used for temporary and permanent attachment may be of less importance” in page 10 lines 331-332.

Yes, the villi on the attachment disc are believed to play a key part in temporary and permanent adhesion of barnacle cyprids, if so, the group 4 cyprids have the lowest villus density among the species studied here, suggesting that the conventional chemical mechanisms used may be of less importance.

Comment: *Line 333. What do you mean by “the function of these extensions remain”?*

Response: We have rephrased as “The function of these calcareous extensions remain unknown” in page 10 lines 347-348.

In our observation, it seems that *E. dofleini* juveniles used the extending calcareous structures to anchor host sponge rather than cement adhesion.

Referee 4:

Comment: *I found this to be an interesting, novel and well-constructed contribution to the literature on barnacle biology, and larval settlement ecology more broadly. I would urge the authors to have the paper proof-read, since the grammar and sentence structure, including mixed tenses etc., let it down. It is nice work and deserves to be seen in the best light.*

I am reviewing the most recent version of the paper, of course, but it appears that the comments of previous reviewers (the reasonable ones, at least) have been addressed.

April 03, 2020

I have one significant query. The fact that the authors are so confident about their terminology, and that it was not raised in previous reviews, could suggest that this is my inadequate understanding, but I would ask for some clarity on use of the definition 'symbiont' or 'symbiosis' to describe the relationship between barnacles and their sponge hosts. This seems to be more of a parasitic relationship to me (what does the sponge get out of it?) or at least epi/endo-biotic. Lines 60-63 compounded my views on this. Surely cooperation is a hallmark of a symbiotic relationship, rather than challenging or deterring the other party? I ask for either a change in terminology or, in the introduction, a statement defining how the relationship is a symbiotic one.

Response: Thank you for your positive comments. The definition of symbiosis is often broad, it generally indicate species living together is symbiosis. Usually it involve the symbiont and its hosts.

Symbiosis can be divided into different categories as commensalism, mutualism and parasitism, according to whether the organisms involved benefit or suffer from the relationship (Castro and Huber, 2010; Simon et al., 2016; Dimijian, 2000.). In the introduction (page 3, lines 77-87) we stated “Previous studies proposed the relationship between Sponge-associated barnacles and the sponges is mutualism; barnacles can obtain trophic advantages and gain protection from host sponges. Barnacles inside sponges can strengthen the sponge skeleton and enhance trophic intake of host sponge [Ilan et al., 1999]. There are however, studies suggested the barnacle and sponge relationship as commensalism, that barnacles obtain protection from host sponge and/or further reduce investment to shell physical armor [Kolbasov, 1993, Ilan et al., 1999 and Magnino et al., 1999] and the host neither suffers nor gains by housing the barnacles inside. Until now, the precise relationship between barnacles and their host sponge is uncertain. In this study, we consider the sponge associated barnacles are symbiotic (living together) with their hosts without further evaluating their specific relationships due to lack of evidence at this moment.” We hope the reviewer can agree our decision and rationale on keeping the use of the term symbiotic for the relationship between barnacles and sponges without further evaluation.

Simon EJ, Dickey JL, Reece JB. 2016. Essential biology. United States of America, Pearson Education, Inc.

Castro P, Huber ME. 2010. Marine Biology (8th eds), McGraw-Hill Publishing Company, Inc.

Dimijian GG. 2000. Evolving Together: The Biology of Symbiosis, Part 1, Baylor University Medical Center Proceedings, 13, 217a-226 (doi.org/10.1080/08998280.2000.11927678)

Ilan M, Loya Y, Kolbasov GA, Brikner I. 1999 Sponge inhabiting barnacles from coral reefs. Mar. Biol. 133, 709–716. (doi:10.1007/s002270050)

Magnino G, Pronzato R, Sara' A, Gaino E. 1999 Fauna associated with the horny sponge *Anomoianthella lamella* Pulitzer-Finali & Pronzato, 1999 (Ianthellidae, Demospongiae) from Papua-New Guinea. Ital. J. Zool. 66, 175–181.

April 03, 2020

(doi:10.1080/11250009909356253)

Kolbasov GA. 1993 Revision of the genus *Acasta* leach (cirripedia: balanoidea). Zool. J. Linn. Soc. 109, 395–427.

(doi.org/10.1111/j.1096-3642.1993.tb00307.x)

We hope that these changes render our manuscript suitable for publication in *Proceedings of the Royal Society B*.

Sincerely,

Meng-Chen Yu, Niklas Dreyer, Gregory A. Kolbasov, Jens T. Høeg and Benny K.K. Chan